# *Bacillus**velezensis* Strains for Protecting Cucumber Plants from Root-Knot Nematode *Meloidogyne incognita* in a Greenhouse

**DOI:** 10.3390/plants11030275

**Published:** 2022-01-20

**Authors:** Anzhela M. Asaturova, Ludmila N. Bugaeva, Anna I. Homyak, Galina A. Slobodyanyuk, Evgeninya V. Kashutina, Larisa V. Yasyuk, Nikita M. Sidorov, Vladimir D. Nadykta, Alexey V. Garkovenko

**Affiliations:** 1Federal Research Center of Biological Plant Protection, p/o 39, 350039 Krasnodar, Russia; biocontrol-vniibzr@yandex.ru (A.M.A.); elisitor@mail.ru (N.M.S.); vnadykta@mail.ru (V.D.N.); 2Lazarevskaya Experimental Plant Protection Station, the Affiliated Branch of the Federal Research Centre of Biological Plant Protection, l. 200, Sochi Highway-77, 354200 Sochi, Russia; bugaevaln@mail.ru (L.N.B.); bugaeval@mail.ru (G.A.S.); kashutinaev@mail.ru (E.V.K.); gnu_oszr@mail.ru (L.V.Y.); 3Shemyakin-Ovchinnikov Institute of Bioorganic Chemistry of the Russian Academy of Sciences, Miklukho-Maklaya Str. 16/10, 117997 Moscow, Russia; garkovenko@gmail.com; 4Laboratory of Molecular Genetic Research in the Agroindustrial Complex, Department of Biotechnology, Biochemistry and Biophysics, Trubilin Kuban State Agrarian University, Kalinina Str. 13, 350044 Krasnodar, Russia

**Keywords:** *B. velezensis*, cultivation conditions, cucumber, root-knot nematode, *Meloidogyne incognita*, greenhouse

## Abstract

*Meloidogyne incognita* Kofoid et White is one of the most dangerous root-knot nematodes in greenhouses. In this study, we evaluated two *Bacillus* strains (*Bacillus velezensis* BZR 86 and *Bacillus velezensis* BZR 277) as promising microbiological agents for protecting cucumber plants from the root-knot nematode *M. incognita* Kof. The morphological and cultural characteristics and enzymatic activity of the strains have been studied and the optimal conditions for its cultivation have been developed. We have shown the nematicidal activity of these strains against *M. incognita*. Experiments with the cucumber variety Courage were conducted under greenhouse conditions in 2016–2018. We determined the effect of plant damage with *M. incognita* to plants on the biometric parameters of underground and aboveground parts of cucumber plants, as well as on the gall formation index and yield. It was found that the treatment of plants with *Bacillus* strains contributed to an increase in the height of cucumber plants by 7.4–43.1%, an increase in leaf area by 2.7–17.8%, and an increase in root mass by 3.2–16.1% compared with the control plants without treatment. The application of these strains was proved to contribute to an increase in yield by 4.6–45.8% compared to control. Our experiments suggest that the treatment of cucumber plants with two *Bacillus* strains improved plant health and crop productivity in the greenhouse. *B. velezensis* BZR 86 and *B. velezensis* BZR 277 may form the basis for bionematicides to protect cucumber plants from the root-knot nematode *M. incognita*.

## 1. Introduction

In greenhouses, the diversity of harmful species is less than in the field, however, the specific microclimate in greenhouses, as well as the absence of natural enemies as regulatory factors, lead to pest accumulation and increase its harmfulness to cultivated plants. Parasitic phytonematodes are among the most serious pests that negatively affect the quality of vegetables in a greenhouse [1,2]. They are obligate parasites that feed mostly on plant roots with common aboveground symptoms of stunting, yellowing, wilting, and yield losses and belowground root malformation due to direct feeding damage. These factors lead to yield losses [3]. Parasitic phytonematodes feed on many crops worldwide and they can cause enormous yield losses with an estimation of 100 billion dollars a year [4].

Root-knot nematodes are one of the most harmful groups of phytophagous organisms in greenhouses. About 5% of world crop production is annually destroyed by *Meloidogyne* spp. Gall nematodes induce overgrowth of root cells, which leads to the formation of galls on plant roots. The nematode damages the vascular tissue of the roots, thereby interfering with the normal movement of water and nutrients. Infected plants show signs of nutrient deficiency: slow growth, yellowing of leaves, wilting, and dying off of the plant. In such a condition, the yield of plants can drop by 50–80%, and in some cases, there is an entire yield loss [5]. Control of nematodes in greenhouses is expensive and time-consuming, and none of the existing methods completely relieves the plant from its presence. To reduce the number of nematodes in greenhouses, agricultural producers use resistant varieties and steam the soil. In addition, soil fumigation with highly toxic substances prohibited in greenhouses is often used. [6].

In recent years, there have been numerous studies worldwide concerning the possibility of reducing the harmfulness of phytopathogenic nematodes using antagonistic microorganisms [7,8,9]. Active strains of antagonistic bacteria with high nematicidal activity in combination with high biological and economic efficiency have been identified. Special attention of the experts is paid to *Bacillus*, *Pseudomonas*, and *Pasteuria* bacterial strains [10,11,12,13]. In addition, many researchers have found the nematicidal activity of the metabolites of *Trichoderma*, *Paecilomyces*, *Arthrobotrys*, *Beauveria*, *Pochonia*, *Fusarium*, and *Myrothecium* fungi [14,15,16].

Currently, successful research has been carried out and technologies have been developed for the development of biological products, such as Bioact WG (*Purpureocillium lilacinus*), KlamiC (*Pochonia chlamydosporia*), Econem (*Pasteuria penetrans*), Deny, Blue Circle (*Burkholderia cepacia*), DiTera (*Myrothecium Verruotia*), and Nortica VOTiVO (*Bacillus firmus*). Giant corporations, such as BASF and Bayer CropScience, which are key suppliers of plant protection products in the world market, became interested in the production of these preparations. It should be noted that some products are produced only for use in the country of manufacture [17]. However, none of these biological products are registered for use in Russia.

Currently, only one formulation, Phytoverm, based on *Streptomyces avermectilis* metabolites, is registered against phytonematodes in Russia. However, this group of products belongs to the category of agrochemicals and is prohibited for use in organic farming technologies. So far, there is no registered biological product based on microorganisms against phytonematodes in Russia [18]. The reasons for such a poor assortment of biological products are the insufficient knowledge of the biological characteristics of bacterial and fungal strains as the basis of nematicidal biological products, as well as the study of their trophic needs and resistance to various factors.

Thus, one of the promising solutions to the problem of protecting plants from *M. incognita* can be biological control of the pest population based on the use of effective microbiological agents that are currently not available in Russia.

This study focused on the isolation, identification, and characterization of *B. velezensis* BZR 86 and *B. velezensis* BZR 277 isolated from winter wheat roots. They showed high fungicidal activity against *F. graminearum* [19] and high nematicidal activity in vitro against the root-knot nematode *M. incognita* Kofoid et White [20]. The strains were selected from the BRC «State collection of beneficial insects, mites and microorganisms» of the Laboratory for the creation of microbiological plant protection products and the collection of microorganisms (FSBSI FRCBPP, Russia), since they have enzymatic, growth-stimulating, and nematicidal activity against the root-knot nematode *M. incognita*. In addition, the conditions for cultivating *Bacillus* strains were optimized.

## 2. Results

### 2.1. Characteristics and Identification of Bacillus strains

The cells of the *B. velezensis* BZR 86 strain are rod-shaped with rounded ends; single or paired; cells are motile; and have sizes of 2.8–4.1 × 6.6–9.4 microns. The cells usually contain spores. Coloring indicates positive results in the Gram stain. On the MPA, the shape of the colonies is round with a scalloped edge. The colonies are shiny and colorless. The profile of the colonies is flat, the edge is wavy, the structure is fine-grained, the texture is soft, and the colonies do not stick to the loop. The diameter of the colonies is 2–4 mm. On the PGA, rhizoid colonies with a smooth edge are formed. The colonies are matte and cream-colored. The profile of the colonies is convex, the structure is streaky in the center, fine-grained along the edge, the texture is soft, mucous, and the colonies adhere to the loop. The diameter of the colonies is 1–3 mm.

*B. velezensis* BZR 277 cells are rod-shaped with rounded ends; single or paired; cells are motile; and have sizes of 1.3–1.8 × 4.5–5.4 microns. The cells usually contain spores. Coloring indicates positive results in the Gram stain. On the MPA, the shape of the colonies is round with a scalloped edge. Colonies are opaque and colorless. The profile of the colonies is flat, the structure is fine-grained, the texture is soft, and the mucous colonies adhere to the loop. The diameter of the colonies is 0.5–2 mm. On the PGA, rhizoid colonies with a smooth edge are formed. Colonies are matte and cream-colored or yellowish-brown. The profile of the colonies is curved, the structure is streaky, the consistency is soft, mucous, and the colonies adhere to the loop. The diameter of the colonies is 1–3 mm.

MALDI-TOF MS analysis of bacterial colonies of the studied strains showed score values of ≥0.65 (BactoSCREEN ID) and ≥1.952 (Bruker Autoflex). These data confirm taxonomic identification only up to the genus Bacillus (Figure 1).

A phylogenetic tree based on genome-wide sequencing of 120 conserved marker genes shows that strains BZR 86 and BZR 277 clearly cluster with *B. velezensis* NRRL B-41580 (*B. velezensis* GCF 001461 825.1 at the phylogenetic level); the average nucleotide identity (ANI) is 97.59% (Figure 2). Strains BZR 86 and BZR 277 are closer to *B. siamensis* and B*. amyloliquefaciens* than to *B. subtilis* (as determined using 16S rRNA).

### 2.2. Enzymatic Activity of B. velezensis BZR 86 and B. velezensis BZR 277 Strains

It is known that effective lysis of the cell walls of pests is associated with the complex action of various hydrolytic enzymes. Therefore, the ability of *Bacillus* strains to produce various hydrolytic enzymes was studied. We revealed a different level of synthesis of lytic enzymes in the bacterial strains (Table 1).

The *B. velezensis* BZR 277 strain showed the ability to synthesize protease and lipolytic enzymes, while *B. velezensis* BZR 86 strain showed the ability to synthesize chitinases only. Both strains produce gelatinase. When inoculating with a prick on a gelatinous medium in a test tube, it was noted that the *B. velezensis* BZR 86 strain forms a crater-like liquefaction of the gelatinase medium, and the *B. velezensis* BZR 277 strain forms a turnip-like one.

### 2.3. Cultivation Conditions for B. velezensis BZR 86 and B. velezensis BZR 277 Strains

In our studies, five parameters were identified that affect the growth of strains in the process of periodic cultivation: cultivation temperature, the acidity of the medium, sources of carbon and nitrogen nutrition, and cultivation time. During the study, we found that some parameters, such as temperature and acidity of the environment, have a significant effect on cell growth. When the incubation temperature changed from 20 to 35 °C, and the acidity of the medium from 3 to 10, there was significant variation in the titer (Table 2).

A high spore or cell titer for the *B. velezensis* BZR 86 strain was noted at a temperature of 35 °C and for the *B. velezensis* BZR 277 strain at a temperature of 30 °C. The optimal pH for both strains was three. Different nutrient sources had a more significant effect on cell growth. Thus, molasses had appeared to be the optimal carbon source for both strains. The highest titer was recorded on a nutrient medium, where peptone was used as a nitrogen source. The maximum cell titer for both strains was noted in the range of 24–36 h of cultivation. When studying the influence of the cultivation time on the dynamics of the growth of the strains, it was noted that the phase of initial growth (the period when the volume of cells increased, but not their number) began after the introduction of the parent culture into the medium and lasted up to 8 h. The phase of the most active growth began 16 h after the introduction of the parent culture and reached its maximum in 24–36 h for both strains. This was followed by a phase of withering away.

### 2.4. Study of B. velezensis BZR 86 and B. velezensis BZR 277 Strains under Greenhouse Conditions

As a result of comprehensive monitoring, it was found that the cause of the formation of the galls on plant roots is the root-knot nematode M. incognita. In 2016–2018, *B. velezensis* BZR 277 and *B. velezensis* BZR 86 strains were tested against the M. incognita on cucumber plants in the greenhouse. In 2016, a single soil treatment under the root during planting with the studied strains led to plant growth retardation, shredding, and necrosis of the leaves, and then plants wilting. The double treatment under the root with Bacillus strains during the entire growing season caused intensive plant development and no visual signs of plant damage by the nematode were observed. Growth and development indicators of plants after treatment with Bacillus strains significantly exceeded the control. The vegetative period of plants lasted for three months. To the end of the period, complete drying of the control plants was noted. Phytopathological analysis of cucumber plant samples showed that no signs of damage by peronosporosis and powdery mildew were discovered on the plants (Figure 3).

In 2016, the gall index for the *B. velezensis* BZR 86 strain was five times less, and for the *B. velezensis* BZR 277 strain was 14 times less, than in the control. However, yield is provided not only by a decrease in the number of galls, but also by a direct effect on the growth and development of plants. Therefore, the yield of cucumber plants treated with the studied strains was 5.2% higher for *B. velezensis* BZR 277 (a double treatment), and 21.4% higher for *B. velezensis* BZR 86 (a double treatment) compared to the control (Figure 4).

In 2017, a double treatment of cucumber plants with *B. velezensis* BZR 86 and *B. velezensis* BZR 277 strains was statistically more effective than a single treatment. It was noted that the *B. velezensis* BZR 277 strain had a greater effect on the development of the aerial part of cucumber plants, while the *B. velezensis* BZR 86 strain stimulated the development of the root system (Figure 3 and Figure 5). Due to the increase in the underground part, a higher percentage of nematode damage was noted for *B. velezensis* BZR 86 strain: two times compared to *B. velezensis* BZR 277. Both strains were noted to influence an increase in yield by 17.8–45.8% compared to the control (Figure 4).

In 2018, a double treatment of cucumber plants with the *B. velezensis* BZR 86 and *B. velezensis* BZR 277 strains also turned out to be more effective than a single treatment. This may indirectly indicate the ability of the studied strains to enhance the induced systemic resistance of plants. The maximum biometric parameters were recorded for plants treated with the *B. velezensis* BZR 277 strain. It also showed greater activity against M. incognita when it was applied to the root system; two times fewer galls were noted compared to the strain B velezensis BZR 86. As a result, the maximum cucumber yield was recorded with double application of the *B. velezensis* BZR 277 strain—32.7% more than in the control, and 12.7% more than in the option with a double application of the *B. velezensis* BZR 86 strain (Table 1).

## 3. Discussion

In this study, we assessed the disease control efficacy of *B. velezensis* BZR 86 and *B. velezensis* BZR 277 strains against the root-knot nematode *M. incognita* of cucumber plants in the greenhouse.

Many researchers note the ability of *Bacillus* strains bacteria to reduce the number and harmfulness of phytoparasitic nematodes that attack agricultural crops [3,21]. One of the mechanisms of the nematicidal action of Bacillus strains is the synthesis of extracellular enzymes [22,23,24,25]. It is likely that extracellular enzymes, such as proteases, lipases, and chitinases, are capable of destroying the physical and physiological integrity of the nematode cuticle and eggs [26]. It was shown that enzymes can damage the egg membrane of nematodes, which consists of a protein matrix and a chitinous layer [27]. In addition, chitin is the main component of the nematode pharynx. Therefore, any disturbance in the synthesis or hydrolysis of chitin can lead to the death of nematode embryos, the laying of defective eggs, or a violation of molting [28]. The enzymatic activity (chitinase and protease) of the *B. megaterium* culture supernatant provided a nematicidal effect in the range of 21–30% against the larvae of *Meloidogyne* sp. and 30–37% against its eggs [29].

Some studies revealed the ability of Bacillus firmus to synthesize toxins that reduced the number of eggs of *Meloidogyne* spp. [30]. It is known that effective lysis of the cell walls of pests is associated with the complex action of various hydrolytic enzymes. Geng et al. [31] found the ability of *B. firmus* to produce peptidase group enzymes capable of destroying intestinal tissue of nematodes. That research served as the basis for the development of biological products Nortica and VOTiVO, which were successfully used in the United States. Other studies revealed 100% mortality of the second age larvae of the root-knot nematodes *M. incognita* and *M. javanica* within 72 h caused by liquid cultures based on *B. thuringiensis* and *P. fluorescens* bacterial strains. Furthermore, these bacteria stimulated plant growth [32].

Our results demonstrated the activity of *Bacillus* strains against *M. incognita* on cucumber plants (up to 76.4%). According to the study, we assumed the ability of Bacillus strains to synthesize proteolytic enzymes that caused the death of nematodes [10]. In Iran, the treatment of tomato plants with the filtrate of the bacterial culture *Bacillus* sp. in vitro caused mortality of *M. incognita* juveniles by 72% [33]. The use of chitinase and protease synthesized by the *B. pumilus* L1 strain in vitro reduced the hatching of *M. arenaria* eggs by 78%, and the mortality of juveniles was 98.6% compared to the control. In this study, structural damage of nematode bodies and eggs was noted [34]. Under in vitro conditions, *B. subtilis* culture filtrates inhibited the hatching of M. incognita eggs by 94.6% and caused mortality of juveniles by 91.3% [35]. Laboratory studies of *B. subtilis* strains exhibiting nematicidal activity against *M. incognita* showed that the use of a cell suspension provided the mortality of juveniles at the level of 39.3%, while the mortality rate when using the supernatant was 82.3–90.7%, which may be caused by the ability of *B. subtilis* to produce lytic enzymes [36]. These data are confirmed by the studies carried out in 2020, during which liquid culture filtrate caused the death of 90% of juveniles of the second stage and reduced egg hatchability by 97% in vitro [37].

Our studies confirmed the ability of *B. velezensis* BZR 86 and *B. velezensis* BZR 277 to produce a number of enzymes. The *B. velezensis* BZR 86 strain discovered chitinolytic activity. As to the *B. velezensis* BZR 277 strain, a high level of synthesis of lipases and proteases was noted, which may be partially involved in the process of suppression of the nematode *M. incognita*. Various factors have a significant effect on the growth of the number of cells and the synthesis of enzymes: temperature, acidity, the composition of the nutrient medium, and the incubation period [38]. Therefore, it is important to optimize the cultivation conditions for improving the nematicidal activity of bacterial strains. For example, Cheba et al. [39] showed that incubation of the *Bacillus* sp. R2 at 30 °C led to the highest level of chitinase synthesis. *B. cereus* JK-XZ3 filtrate cultivated at 30 °C showed the highest nematicidal activity, resulting in 100% mortality of Bursaphelenchus xylophilus [40]. However, Dukariya and Kumar [41] reported maximum chitinase productivity at 37 °C.

In our studies, we improved the cultivation conditions for strains exhibiting nematicidal activity according to a criterion, such as the number of colony-forming units. It is known that *Bacillus* strains are capable of growing in a temperature range of 5.5 °C to 55.7 °C [42,43,44]. According to the data obtained for the *B. velezensis* BZR 86 strain, the optimum temperature is 35 °C, and for the *B. velezensis* BZR 277 strain—30 °C. Our study is fully consistent with the conclusions of Park et al. [45], according to whom, when cultivating the B. velezensis GH1–13 strain, a titer of 7.5 × 10^9^ CFU/mL was obtained at 37 °C. A similar pattern was noted in the study by Usanov et al. [46], according to which an increase in the cultivation temperature to 40 °C has a positive effect on the growth dynamics of the *B. subtilis*, and the number of colony-forming units exceeds the control by 88.9%.

In our study, the maximum titer of both strains was observed at pH 3. However, it should be noted that when the strains were cultivated on media with different levels of acidity, no significant jumps in cell density were observed. These data are confirmed by Mohapatra et al. [47] 2015, according to whom, insignificant fluctuations in CFU were observed during cultivation of the *Bacillus* sp. P1on media with pH 6–9 (5.0–8.4 × 10^5^ CFU/mL). *B. velezensis* XT1 strain was able to grow in a wide pH range of 5–10 [48]. A relatively high cell density at pH 3 was noted during the cultivation of the *B. velezensis* CE 100 strain, the initial pH of the medium of 7.5 decreased to 4.7 over 24 h of cultivation, which may indicate the ability of some *B. velezensis* strains to grow in an acidic medium [49].

Molasses has been shown to optimize the parameters of growing strains-producers of biological products [50,51]. Being a source of not only sugars, but also vitamins, macro- and microelements, this ingredient ensures the active growth of microorganism cultures. For strains *B. velezensis* BZR 86 and *B. velezensis* BZR 277, the cell titer in liquid cultures using molasses as media component was 2–3 times higher than on media with sucrose, glucose, and glycerol and amounted up to (1.6 ± 0.03) × 10^9^ CFU/mL for the *B. velezensis* BZR 86 strain and (5.8 ± 0.39) × 10^8^ CFU/mL for the *B. velezensis* BZR 277 strain. Peptone was the most preferred source of nitrogen nutrition for both strains. It should be noted that the B. velezensis BZR 86 strain turned out to be more sensitive to the source of nitrogen nutrition: when NaNO_3_ was added to the nutrient medium, the titer was (3.7 ± 0.4) × 10^6^ CFU/mL, and when peptone was added—(1.8 ± 0.07) × 10^8^ CFU/mL. These results are consistent with the studies on the optimization of the nutrient medium for the cultivation of the *B. velezensis* NRC-1 strain, according to which the maximum cell growth and mannanase synthesis were achieved on the medium with peptone [52].

The cultivation time is an important parameter for the growth of biocontrol agents. If bacterial cultures are incubated for too long, some metabolites may be converted to other compounds. In contrast, if the incubation period is not long enough, it is possible that important metabolites have not yet been synthesized (for example, enzymes that are formed during the stationary growth phase). This demonstrates the importance of making bacterial growth curves [53]. During our research, it was noted that the phase of initial growth began after the introduction of a parent culture with a titer of (1.4 ± 0.05) × 10^8^ CFU/mL in the *B. velezensis* BZR 277 strain and (8.1 ± 0.01) × 10^7^ CFU/mL in the *B. velezensis* 86 strain and lasted up to 8 h. The phase of the most active growth began 16 h after the introduction of the mother culture and began to decay after 36 h for the *B. velezensis* 277 strain and from 8 to 36 h for the *B. velezensis* 86 strain. The maximum cell titer was observed during this period and amounted to (1.7 ± 0.02) × 10^9^ CFU/mL in the *B. velezensis* BZR 277 strain and (1.3 ± 0.11) × 10^9^ CFU/mL in the B. velezensis BZR 86 strain. The phase of bacterial cell death occurred 36 h after the start of cultivation for both strains, which contradicts the data obtained in the study of the cultivation of the *B. velezensis* IP22 strain, according to which the stationary phase lasted up to 60 h [54].

Three-year vegetation tests on the Courage variety of cucumber plants under the greenhouse conditions showed that plants treated with the B. velezensis BZR 86 and *B. velezensis* BZR 277 strains developed more intensively; the growth and development indicators significantly exceeded the control ones. This tendency was observed in research by Sahebani and Omranzade [55], according to which the introduction of a liquid culture based on *B. megaterium* wr101 into the soil infected with *M. javanica* contributed to an increase in the mass of cucumber shoots twice as compared to the control. Other authors showed that the decrease in the number of M. incognita contributed to the improvement of plant development, which was manifested by an increase in biometric indicators and in cucumbers compared with the control. It was noted that the use of two Bacillus strains contributed to the formation of an additional cucumber yield from 4.6% to 45.8%, which confirms the results of the 2019 studies, in which the treatment of cucumber plants with a liquid culture based on the B. subtilis Bs-1 strain provided an additional yield of up to 53.1% in combination with a high nematicidal effect against *M. incognita* [56].

In our research, we showed that in three consecutive years, application of two Bacillus strains in the greenhouse demonstrated that the number of galls per gram of cucumber root mass decreased by 13 times for the *B. velezensis* BZR 277 strain, and by 7.2 times for the *B. velezensis* BZR 86 strain. In general, the *B. velezensis* BZR 277 strain exhibited a more pronounced nematicidal and plant growth-promoting effect. Probably, such difference is due to the selectivity of the nematicidal action of lipases and proteases synthesized by the *B. velezensis* BZR 277 strain against predominantly adult nematodes. It should be noted that the cuticle of nematodes contains different types of structural proteins and their proportions change throughout their life cycle as other authors have shown [54,57].

## 4. Materials and Methods

### 4.1. Microorganisms

*B. velezensis* BZR 86 and *B. velezensis* BZR 277 were isolated from the field soil of the Krasnodar region (45°1′ N, 38°59′ E). Winter wheat plants were selected under aseptic conditions, placed in sterile bags, and stored in a refrigerator at +4 °C. The strains were isolated from the root zone of winter wheat plants by the method of successive washing of the roots from adjacent soil particles [58]. The roots with soil were placed in a flask with 100 mL of sterile water and shaken for 40 min at 180 rpm. The roots were removed from the flask with sterile tweezers and transferred to the next flask containing 100 mL of sterile tap water. The procedure was repeated, successively washing the roots in three flasks (40 min each). The bacteria were inoculated by streaking from the second and third flasks. The *B. velezensis* BZR 277 strain was inoculated on potato-glucose agar and the *B. velezensis* BZR 86 strain on a medium with chitinase. Petri dishes with *B. velezensis* BZR 86 strain were incubated at +28 °C for three days. Petri dishes with *B. velezensis* BZR 277 strain were incubated at +4 °C for 21 days. Isolated colonies were streaked onto potato-glucose agar. This procedure was repeated until a pure culture of the strain was obtained in the Petri dish.

Strains deposited in the Bioresource collection “State collection of beneficial insects, mites and microorganisms” of the Federal State Budgetary Scientific Institution “Federal Research Center of Biological Plant Protection” (FSBSI “FRCBPP”) (registry number 585858).

### 4.2. MALDI-TOF MS Analysis

MALDI-TOF MS analysis was performed to identify bacterial cultures. The taxonomic status of bacterial cultures was determined by the BactoScreen analyzer. The spectra were analyzed using the BactoScreen-ID software version 2.4. In addition, the BRUKER autoflex III smartbeam and flexcontrol 3.0 software was used. The analysis was carried out using a database containing the spectra of 3995 microorganisms. The values obtained were expressed as a logarithmic score. In particular, a score of 2.0—identification at the species level is allowed, a score in the range of 1.7 ± 2.0—identification only at the genus level, a score below 1.7—the absence of significant similarity of the obtained spectrum with any record of the database (not reliable identification). *E. coli* DH5a proteins were used as a bacterial standard.

### 4.3. Molecular Genetic Identification of Strains

Isolation of genomic DNA preparations of *B. velezensis* BZR 86 and *B. velezensis* BZR 277 strains was performed using the DNeasy PowerSoil Kit, QIAGEN (Hilden, Germany) according to standard protocols. The amount of isolated DNA was determined by the fluorometric method using Qubit dsDNA HS Assay Kit, ThermoFisher Scientific (Waltham, MA, USA) according to the manufacturer’s protocols.

To determine the complete genomes of *B. velezensis* BZR 86 and *B. velezensis* BZR 277 strains, a combined strategy was used, including the use of two high-productive sequencing platforms—Illumina (MiSeq) and monomolecular sequencing on MinIon (Oxford Nanopore). At the first stage, a genomic library of “random fragments” was prepared, suitable for sequencing on the MiSeq device (Illumina) using the NEBNext ultra II DNA Library kit (NEB) and then read on the MiSeq genomic analyzer. At the second stage, the genome was additionally sequenced using monomolecular nanopore sequencing technology (MinION instrument from Oxford Nanopore). To prepare genomic libraries suitable for sequencing on the MinION device, a Ligation Sequencing kit 1D (Oxford Nanopore) was used according to the manufacturer’s recommendations. Sequencing on the MinION was performed using the Ligation Sequencing kit 1D protocol using FLO-MIN110 wells. The sequencing results were saved in a FAST5 file. Using the flash program [59], paired intersecting reads obtained on MiSeq (Illumina) were combined and the poor-quality ends of the reads were cut using the Sickle program. Structural (protein-coding) genes and ribosomal RNA genes were identified and their functions were theoretically predicted using the RAST server [60].

Multiple alignments of the concatenated amino acid sequences of 120 bacterial single-copy marker genes were performed using the Genome Taxonomy Data Base (GTDB-Tk v. 1.3.0 toolkit software) from RefSeq and Genbank genomes (USA) [61]. This multiple alignment was used to construct a maximum similar phylogenetic tree using PhyML v.3.3 with default parameters [62]. Internal branching support was assessed using a Bayesian test in PhyML.

The obtained *B. velezensis* BZR 86 sequences were deposited in the NCBI database under accession numbers PRJNA677970 (BioProject), SRX9502286 (SRA), and SAMN16784691 (BioSample). The obtained *B. velezensis* BZR 277 sequences were deposited into the NCBI database under accession numbers PRJNA588983 (BioProject), SRX9502288 (SRA), and SAMN16784690 (BioSample).

### 4.4. Cultural and Morphological Characteristics of Bacillus strains

Cultural and morphological characteristics of the strains were studied on meat-peptone agar (MPA) and potato-glucose agar (PGA) using an Axio Scope A1 microscope (Jena, Germany). The shape and size of the cells, the ability to form spores, the location of spores in the cells, the ability to move, and coloring were determined according to Gram. We studied the growth characteristics, shape, size, surface, profile, color, shine, and transparency of the colonies, as well as their edge, structure, and consistency [63].

### 4.5. Enzymatic Activity of Bacillus strains

The enzymatic activity of antagonistic bacterial strains was carried out using generally accepted methods [64]. Chitinase, lipase, and protease activity was determined. To determine the chitinolytic activity, a synthetic medium of the following composition was used, g/l: sucrose—20.0; NaNO_3_—3.0; KH_2_PO_4_—1.0; MgSO4—0.3; chalk—10.0; and agar—20.0. The medium was sterilized by autoclaving, poured into Petri dishes, and cooled. Inoculating of bacterial strains was performed by streaking. Petri dishes were incubated for 7–10 days at a temperature of +28.0 °C. Chitinase activity was judged by the formation of clearing zones around the colonies.

Lipolytic activity was determined on yolk agar of the following composition, g/l: peptone—40.0; glucose—2.0; Na_2_HPO_4_—5.0; NaCl—2.0; MgSO_4_ 0.5% solution—2.0 mL; and agar—25.0. 4. The medium was sterilized by autoclaving and cooled to +60.0 °C. The egg shell was disinfected with alcohol and allowed to dry. The egg was broken and the yolk was separated from the egg white. The yolk, in compliance with the rules of asepsis, was transferred into molten agar and stirred until a homogeneous suspension was obtained, which was poured into Petri dishes and left to solidify. Inoculating of bacterial strains was performed by streaking. Petri dishes were incubated for 14 days at a temperature of +28.0 °C. Then the lid of the Petri dish was removed and the surface was carefully examined under oblique illumination. Lipolytic activity was judged by the formation of an oily, glistening, or nacreous layer above and around the bacterial colony on the agar surface.

To determine protease activity, sterile (autoclaved) skim milk was mixed at +50.0 °C with an equal volume of 4% molten aqueous agar. Inoculating of bacterial strains was performed by streaking. Petri dishes were incubated for 14 days. Protease activity was judged by the formation of clearing zones around the colonies.

Gelatinase activity was tested on meat-peptone gelatin, g/l: meat-peptone broth—39.0; and gelatin—150.0. The medium was poured into test tubes, sterilized by autoclaving, and cooled at room temperature. Inoculating of bacterial strains was carried out by injection. The tubes were incubated for 7–10 days at room temperature. The liquefaction of the gelatin was observed visually. The intensity and form of liquefaction were indicated.

### 4.6. Optimal Conditions for the Cultivation of Bacillus strains

To determine the optimal cultivation temperature, the strains were incubated at 20.0, 25.0, 30.0, and 35.0 °C. Czapek medium for bacteria was used as a basis [63]. Carbon sources sucrose, glucose, molasses, and glycerol were added in the test media. In the study of carbon sources nitrogen sodium nitrate served as the unchanged nutrition component. In determining the optimal sources of nitrogen nutrition peptone, NaNO_3_, yeast and corn extracts were tested with glucose as the only (constant) carbon source. To choose the optimal acidity of the medium, the strains were grown on a liquid medium, g/L KCl—0.5, MgSO_4_ × 7H_2_O—0.5, K_2_HPO_4_ × 3H_2_O—1.0, CaCO_3_—3.0, FeSO_4_ × 7H2O—0.01, corn extract—2.0, molasses—20.0 at optimum temperatures. By adding lactic acid or alkali (4N NaOH solution), the medium pH was adjusted to 3.0, 6.0, 8.0, and 10.0 using a Sartorius PB-11 pH-meter (Goettingen, Germany). To determine the optimal cultivation time, samples for analysis were taken after 8, 16, 24, 36, 48, and 72 h from the start of cultivation. All experiments were replicated three times. For all experiments, a liquid culture was obtained by the method of periodic cultivation. Incubation was carried out in thermostat cell cultivation systems (180 rpm) “New Brunswick Scientific Excella E25” (Enfield, CT, USA) for 48 h. Periodic cultivation was carried out in conical flasks (350 mL) with a nutrient medium volume of 100 mL and preliminary introduction of stock culture (2% of the nutrient medium volume). The stock culture was obtained by introducing agar blocks with the studied strains into conical flasks and subsequent cultivation.

At the end of cultivation, the number of bacterial cells was determined by the Koch method in all experiments on MPA [63]. The grown colonies were counted with the Color Qcount, Spiral Biotech, 530 (Canton, MA, USA) system for the automatic counting colonies.

In our research we used Unique Scientific Facility “New generation technological line for developing microbiological plant protection products” of Federal Research Center of Biological Plant Protection, Krasnodar, Russia (http://ckp-rf.ru/%E2%84%96671367, accessed on 21 August 2019).

### 4.7. Greenhouse Evaluation of Bacillus strains

A liquid culture of bacterial strains was obtained in New Brunswick Scientific Excella E25 cell culture systems (180 rpm) on a potato-glucose medium, g/L: potato broth—500.0 and glucose—20.0. Cultivation was carried out for 48 h.

The average temperature in the greenhouse was 27.5 °C in May, 35.4 °C in June, 35.6 °C in July, and 37.5 °C in August.

The tests were carried out in conditions of protected ground in a greenhouse with a total area of 100 m^2^. Five variants of the experiment were randomized in three times repetition and the area of each plot was 5 m^2^. There were eight sample plants in each plot. The cucumber plants of Courage variety with natural infection by root-knot nematode were used. The number of galls was determined by the Guskova method [65]. The starting titer of the liquid culture preparation was 1 × 10^9^ CFU/mL. The application rate of the preparation was 360 L/ha (50–60 mL per plant). The liquid culture was diluted at the rate of 60 mL per liter of water. The consumption rate of the working fluid is up to 6000 L/ha. In total, 200 mL of the preparation was added under the root of each cucumber plant.

Assessment of nematicidal activity of the strains was carried to the scheme:Single treatment by suspension of *B. velezensis* BZR 86 before planting.Double treatment by suspension of *B. velezensis* BZR 86 during planting followed by treatment under the plant root 3 weeks after planting.Single treatment by suspension of *B. velezensis* BZR 277 before planting.Double treatment by suspension of *B. velezensis* BZR 277 during planting followed by treatment under the plant root 3 weeks after planting.Plants without treatment were used as control.

The effectiveness of the bacterial strains was determined by plant height, leaf area, shoot mass and root mass, the ratio of the number of galls to the mass of roots (damage score), and yield.

The root gall severity was based on gall indices (GI) measured on 0 to 5 scales: 0-GI = 0%; 1-GI = 10 to 20%; 2-GI = 21 to 50%; 3-GI = 51 to 80%; 4-GI = 81 to 100%., 5-GI = > 100% on roots [66].

### 4.8. Statistical Analysis

Statistical data processing was performed by standard methods using MS Excel and ANOVA program for Windows. All data were expressed as mean from triplicate samples ± standard deviation. Duncan test was used, and differences were considered statistically significant at *p* < 0.05 level.

## 5. Conclusions

BZR 277 and BZR 86 strains are endophytic bacteria associated with the roots of winter wheat plants from Krasnodar Krai (Russia). They are representatives of *B. velezensis*. They possess the properties of PGPR, exhibiting enzymatic activity and promoting the growth of aboveground and underground parts of the cucumber variety Courage and increasing its yield. In addition, BZR 277 and BZR 86 strains have nematicidal activity against the root-knot nematode *M. incognita*. However, further research is needed on the mechanisms associated with the growth-stimulating and biocontrol activity of the strains.

## Figures and Tables

**Figure 1 plants-11-00275-f001:**
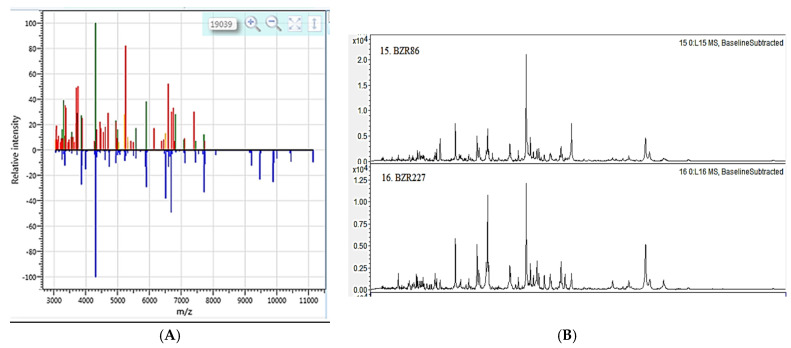
Identification of *Bacillus* strains BZR 86 and BZR 277: (**A**) Unique spectrum of ribosomal proteins of strain BZR 86 (blue peaks belong to the reference strain *Bacillus subtilis*, the rest of the peaks in the upper part of the profile belong to the studied strain BZR 86 (BactoSCREEN)). Green and yellow peaks coincide with the data of the reference strain *B. subtilis*, red ones do not match. Similar data were obtained by this method for the BZR 277 strain (data not shown); (**B**) Unique ribosomal protein spectra of strains BZR 86 and BZR 277 showing taxonomic identity of strains (Bruker).

**Figure 2 plants-11-00275-f002:**
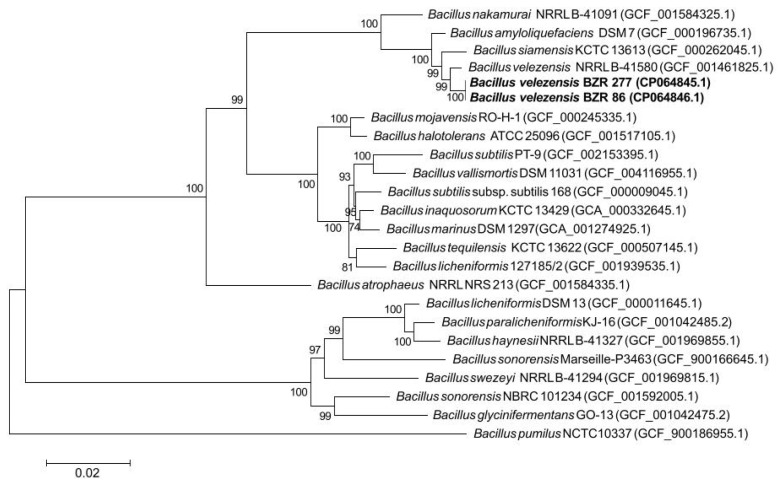
Maximum likelihood phylogenetic tree constructed using amino acid sequences of 120 conserved marker genes. The tree was constructed using PhyML v.3.3.

**Figure 3 plants-11-00275-f003:**
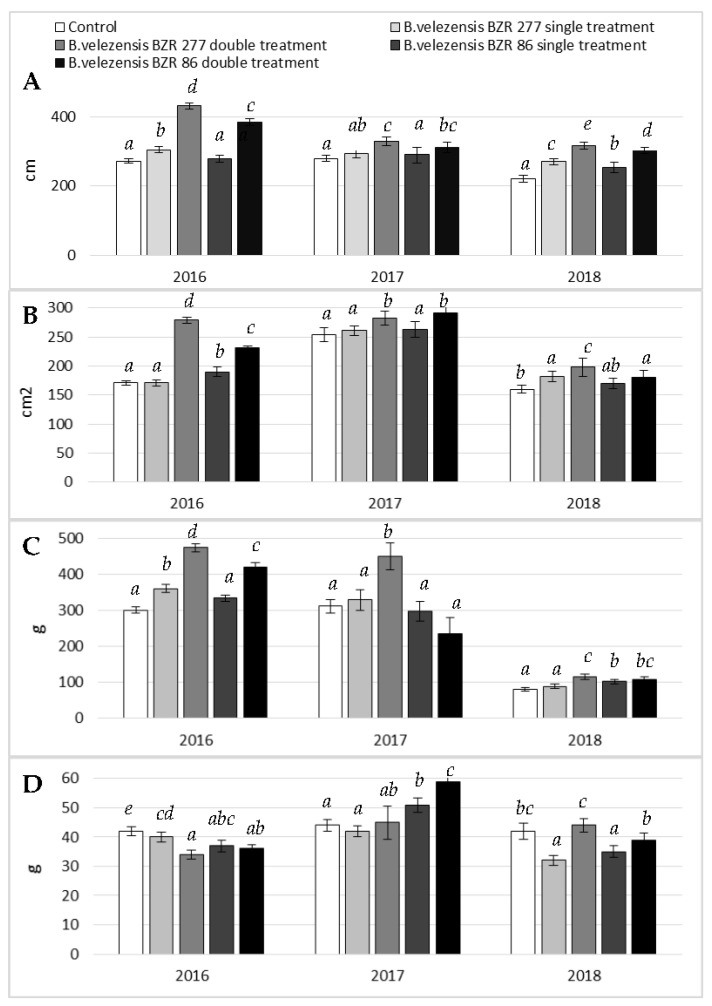
Effect of *B. velezensis* BZR 86 and *B. velezensis* BZR 277 treatment on the growth and development of cucumber plants in the greenhouse, 2016–2018: (**A**) plant height, cm; (**B**) leaf area, cm^2^; (**C**) shoot mass, g; (**D**) root mass, g. The results are presented as the mean ± standard deviation. Different letters in each column indicate significant difference (p ≤ 0.05).

**Figure 4 plants-11-00275-f004:**
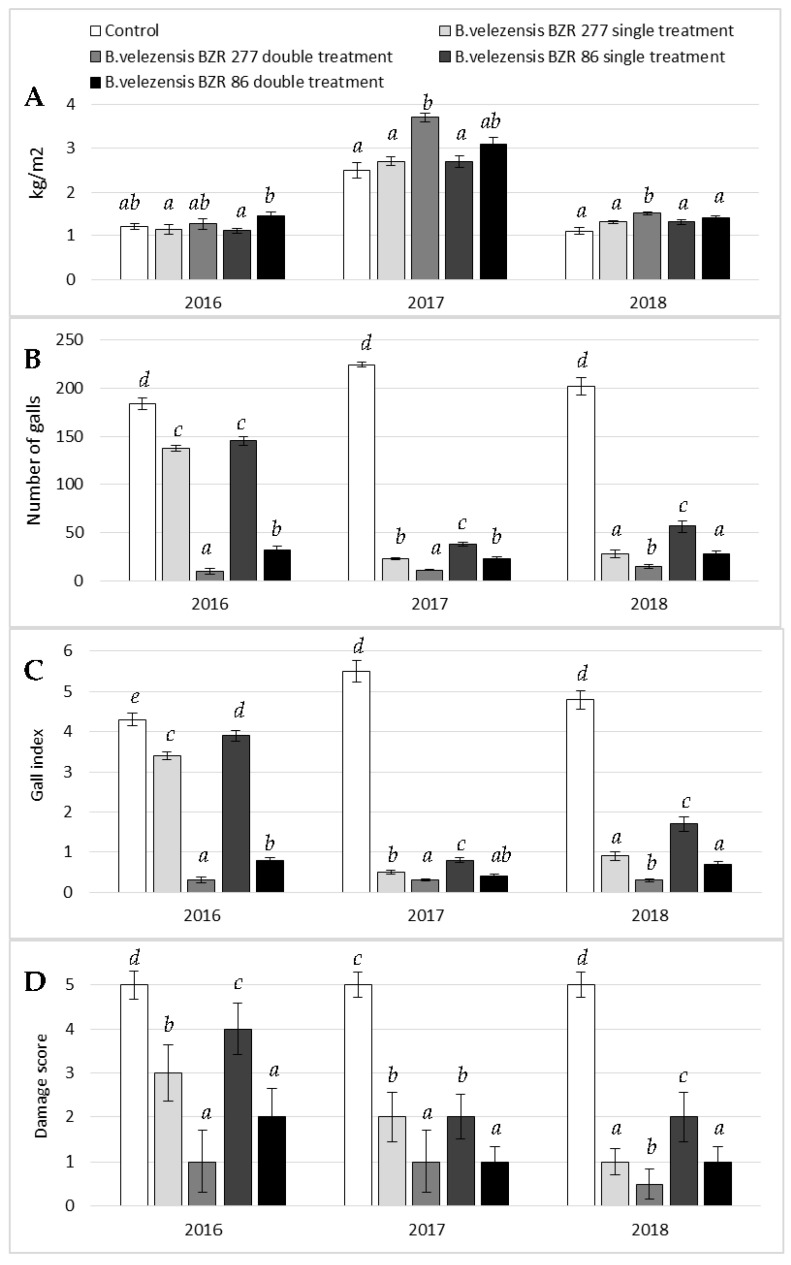
Effect of *B. velezensis* BZR 86 and *B. velezensis* BZR 277 treatment on cucumber yield and gall index in a greenhouse, 2016–2018: (**A**) yield, kg/m^2^; (**B**) number of galls; (**C**) gall index; (**D**) damage score. The results are presented as the mean ± standard deviation. Different letters in each column indicate significant difference (*p* ≤ 0.05).

**Figure 5 plants-11-00275-f005:**
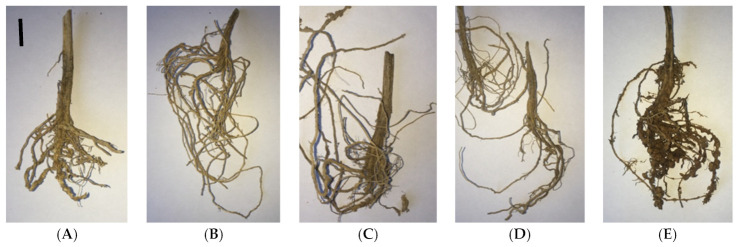
Effect of *B. velezensis* BZR 86 and *B. velezensis* BZR 277 strains on the degree of gall formation on the roots of cucumber plants: (**A**) *B. velezensis* BZR86, single treatment; (**B**) *B. velezensis* BZR86, double treatment; (**C**) *B. velezensis* BZR277, single treatment; (**D**) *B. velezensis* BZR277, double treatment; (**E**) control. Black scale bars represent 1 cm.

**Table 1 plants-11-00275-t001:** Enzymatic activity of *B. velezensis* BZR 86 and *B. velezensis* BZR 277 strains.

Strain	Enzymatic Activity			
	Lipase	Chitinase	Protease	Gelatinase
*B. velezensis* BZR 86	-	+	-	+
*B. velezensis* BZR 277	+++	-	+++	+

- no activity; + low activity; ++ mean activity; +++ high activity.

**Table 2 plants-11-00275-t002:** The number of colony-forming units in liquid cultures is based on strains *B. velezensis* BZR 86 and *B. velezensis* BZR 277, depending on the cultivation conditions.

Parameter	Titer, CFU/ml	
	*B. velezensis* BZR 86	*B. velezensis* BZR 277
Temperature, °C		
20.0	(9.6 ± 0.14) ^1^ × 10 ^6^ b ^2^	(3.4 ± 0.3) × 10 ^5^ b
25.0	(8.6 ± 0.42) × 10 ^6^ a	(6.2 ± 0.14) × 10 ^5^ a
30.0	(8.3 ± 0.67) × 10 ^6^ a	(1.4 ± 0.04) × 10 ^6^ c
35.0	(1 ± 0.05) × 10 ^7^ c	(6.6 ± 0.17) × 10 ^5^ a
pH		
3.0	(3.2 ± 0.06) × 10 ^7^ c	(4.3 ± 0.2) × 10 ^6^ d
6.0	(7.6 ± 0.3) × 10 ^6^ a	(1.7 ± 0.3) × 10 ^6^ c
8.0	(1.1 ± 0.14) × 10 ^7^ b	(1.2 ± 0.02) × 10 ^6^ b
10.0	(1.1 ± 0.4) × 10 ^7^ b	(1.1 ± 0.05) × 10 ^6^ a
Carbon sources		
sucrose	(2.3 ± 0.36) × 10 ^6^ a	(6.2 ± 0.6) × 10 ^5^ a
glucose	(3.1 ± 0.22) × 10 ^6^ a	(6.6 ± 0.75) × 10 ^5^ a
glycerol	(2.3 ± 0.25) × 10 ^6^ a	(1 ± 0.02) × 10 ^6^ a
molasses	(1.6 ± 0.03) × 10 ^9^ b	(5.8 ± 0.39) × 10 ^8^ b
Nitrogen sources		
NaNO3	(3.7 ± 0.4) × 10 ^6^ a	(7 ± 0.66) × 10 ^7^ a
peptone	(1.8 ± 0.07) × 10 ^8^ e	(4.7 ± 0.4) × 10 ^8^ c
yeast extracts	(4 ± 0.2) × 10 ^7^ b	(5.2 ± 0.5) × 10 ^7^ a
corn extracts	(9.4 ± 0.3) × 10 ^7^ c	(1.1 ± 0.05) × 10 ^8^ b
Cultivation time, h		
8	(2.5 ± 0.15) × 10 ^6^ a	(1.2 ± 0.05) × 10 ^7^ a
16	(7.4 ± 0.37) × 10 ^8^ b	(4.7 ± 0.35) × 10 ^8^ b
24	(1.3 ± 0.11) × 10 ^9^ d	(1.7 ± 0.02) × 10 ^9^ d
36	(1.2 ± 0.06) × 10 ^9^ c	(1.6 ± 0.04) × 10 ^9^ c
48	(2.3 ± 0.35) × 10 ^7^ a	(4 ± 0.02) × 10 ^7^ a
72	(1.1 ± 0.2) × 10 ^7^ a	(9.4 ± 0.3) × 10 ^7^ a

^1^ The error corresponds to the standard deviation of three independent analyses. ^2^ Between the options marked with the same letters, when comparing within the columns there are no statistically significant differences according to the Duncan criterion at a 95% probability level-% increase to control. Each optimal parameter is determined while keeping the other parameters unchanged.

## Data Availability

Data is contained within the article.

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
