# Peer review of "Bacillus**velezensis* Strains for Protecting Cucumber Plants from Root-Knot Nematode *Meloidogyne incognita* in a Greenhouse"

_plants, 2022, doi:10.3390/plants11030275_

Round 1

Reviewer 1 Report

accepted

Author Response

Hello!

Thank you for appreciating our manuscript. We have improved the English language. All changes are highlighted in the text. 

Reviewer 2 Report

In this manuscript, the authors isolate two strains of Bacillus from winter wheat root and test the efficiency of anti-nematodes. The aims of this study are clear and the result is interesting to me. I don’t have strong negative comments but have some suggestions below:

  1. For the isolation of bacterial genomic DNA, I am confusing the sources of the commercial kit. The authors use DNeasy PowerSoil Kit (QIAGEN) to extract genomic DNA, but this kit is designed for soil or stool samples. Please check it.
  2. The result of MALDI-TOF MS analysis, is difficult to understand due to a lack of basic information. The authors need to put more information in this section, including the paragraph and figure 1. For example, the authors describe “the blue lines show the reference peaks”. However, I have no idea these reference peaks belong to which taxon or strain. Please revise this section.
  3. The quality of the figure needs to revise. For example, the BZR 86 and B. velezensis BZR 277 strains shown in figure 1 should use bold style or a different color to figure out. 
  4. The information in table3 & 4 can draw figures. It will be easy to compare and understand the results of this study.
  5. The note below table 1 show “– no activity; + low activity; ++ mean activity; +++ high activity”, but does not indicate the mean of “++++” on the column “ B. velezensis BZR 277/ protease”

Author Response

Hello! Thank you for appreciating our manuscript. We have improved the English language. All changes are highlighted in the text.  Your comments are very helpful to us! 

1. For the isolation of bacterial genomic DNA, I am confusing the sources of the commercial kit. The authors use DNeasy PowerSoil Kit (QIAGEN) to extract genomic DNA, but this kit is designed for soil or stool samples. Please check it.

While this kit has been optimized for DNA extraction from soils, it also works well for DNA extraction from bacterial cultures and provides high quality DNA suitable for sequencing.

2. The result of MALDI-TOF MS analysis, is difficult to understand due to a lack of basic information. The authors need to put more information in this section, including the paragraph and figure 1. For example, the authors describe “the blue lines show the reference peaks”. However, I have no idea these reference peaks belong to which taxon or strain. Please revise this section.

We have corrected this part and added the necessary information

3. The quality of the figure needs to revise. For example, the BZR 86 and B. velezensis BZR 277 strains shown in figure 1 should use bold style or a different color to figure out. 

Thank you for your offer. Thanks to him, our article has become better. We did it

4. The information in table3 & 4 can draw figures. It will be easy to compare and understand the results of this study.

Thank you for your offer. Thanks to him, our article has become better. We did it

5. The note below table 1 show “– no activity; + low activity; ++ mean activity; +++ high activity”, but does not indicate the mean of “++++” on the column “ B. velezensis BZR 277/ protease”

Thank you for your comment. Corrected

Reviewer 3 Report

The paper by Asaturova and co-workers is about two Bacillus strains (Bacillus velezensis BZR 86 and Ba-cillus velezensis BZR 277) as promising microbiological agents for protecting cucumber plants from the root-knot nematode M. incognita Kof. The authors studied the morphological and cultural characteristics and enzymatic activity of the strains. They also have developed the optimal conditions for its cultivation.

The article by Asaturova et al. is difficult to read because there are many texts in this manuscript marked in yellow. They are not commented on in any way. It is not known whether they will be deleted or added. Besides, there is no line numbering except section References. The reviewer would have to count them himself. The descriptions under tables 2, 3 and 4 concerning  statistics should be described differently.

In my opinion, this manuscript could be recommended for a publication after taking into account the above doubts.

Author Response

Hello!

Your comments are very helpful to us. 

Changes made earlier are marked in yellow in the manuscript. It will be removed later. Tables 3 and 4 converted into figures.

Reviewer 4 Report

I do find this work interesting and valuable. However, there are some isues which should be addressed before publishing.

Page 4 – Table 1  - protease has four plus signs – is that correct? – Should not be three „+”?

Page 5 in the end and page 6 in the beginning – you wrote that „temperature and acidity of the environment do not have a significant effect on cell growth” but in Table 2 you showed statistically siginificant differences in these parameters.

Page 7 at the beginning, You wrote: „the yield of cucumber plants treated with the studied strains was 5.2 % higher for B. velezensis BZR 277, and 21.4 % higher for B. velezensis BZR86 compared to control” Is this for 2016 and for double treatment? It will be helpful if you specify this.

Page 13 at the bottom – could you describe in a more detailed way this assessment of nematicidal activity: was this area of greenhouse (100 m2) divided into three blocks? I mean: 1.single tratment before planting; 2. Double treatment: first before planting, second 3 weeks after planting (and besides: how this was done exactly – „under the plant root” – what do you mean?); 3. Plants without treatment. In the form as it is in manuscript it is not clear.

What was the number of plots (repetitions)? How the plots were located on this 100m2 area? What was the size of the plot?

What do you mean: „For each treatment, the tests were repeated three times”? Were these: plant height, leaf area, shoot mass and root mass, the ratio of the number of galls to the mass of roots (damage score) and yield measured three times during the growth period? When exactly? Or, there were three repetitions (plots) each year and only one measurement?

Author Response

Hey! Thank you for appreciating our manuscript! 

1. Page 4 – Table 1  - protease has four plus signs – is that correct? – Should not be three „+”?

Thank you for your comment. Corrected

2. Page 5 in the end and page 6 in the beginning – you wrote that „temperature and acidity of the environment do not have a significant effect on cell growth” but in Table 2 you showed statistically siginificant differences in these parameters.

Corrected

3. Page 7 at the beginning, You wrote: „the yield of cucumber plants treated with the studied strains was 5.2 % higher for B. velezensis BZR 277, and 21.4 % higher for B. velezensis BZR86 compared to control” Is this for 2016 and for double treatment? It will be helpful if you specify this.

Thank you for your offer. Thanks to him, our article has become better. We have added the necessary information

4. Page 13 at the bottom – could you describe in a more detailed way this assessment of nematicidal activity: was this area of greenhouse (100 m2) divided into three blocks? I mean: 1.single tratment before planting; 2. Double treatment: first before planting, second 3 weeks after planting (and besides: how this was done exactly – „under the plant root” – what do you mean?); 3. Plants without treatment. In the form as it is in manuscript it is not clear.

What was the number of plots (repetitions)? How the plots were located on this 100m2 area? What was the size of the plot?

What do you mean: „For each treatment, the tests were repeated three times”? Were these: plant height, leaf area, shoot mass and root mass, the ratio of the number of galls to the mass of roots (damage score) and yield measured three times during the growth period? When exactly? Or, there were three repetitions (plots) each year and only one measurement?

100 m2 are divided into five variants: 1 - control without treatment; 2 – single treatment with strain 86; 3 - double treatment with strain 86; 4 - single treatment with strain 277; 5 – double treatment with strain 277.

Five variants of the experiment were randomized in three times repetition, the area of each plot 5m2

Sections were arranged in parallel. Tratment under the plant root - this is watering the soil in the zone of the root system around each plant. Measurements were taken at the end of the growing season every year.

This manuscript is a resubmission of an earlier submission. The following is a list of the peer review reports and author responses from that submission.

Round 1

Reviewer 1 Report

This manuscript describes the characterization and disease control efficacy of two Bacillus strains. Even though the two Bacillus strains effectively suppressed the development of cucumber root-knot nematode diseases in greenhouse, the manuscript was written in poor English and the experimental plan was not designed well.

  1. There is no information why the authors worked suing Bacillus strains.
  2. The objectives are now clear.
  3. There is no story for the two Bacillus strains such as isolation, screening and in vitro nematicidal activity.
  4. The Materials and Methods should be written in details.
  5. As for optimization of cultivation conditions, provide the basal medium and any reference.
  6. Conclusion should be rewritten.
  7. References: There are so many errors.

Lines 2: Delete Kof.

Line 2: Bacillus velezensis strains

Lines 3, 19, 32, 79: against --> from

Line 19, 80, 501: M. incognita

Lines 21&22, 194: M. incognita --> italic

Line 25: Bacillus --> italic

Lines 30-32: Rewrite

Lines 50-53: Rewrite

Lines 58, 60, and 61: The genera should be written in italic.

Line 61: Streptomyces is not fungi.

Lines 60-61: Provide the following references

  • Van Thi Nguyen, Nan Hee Yu, Yookyung Lee, In Min Hwang, Hung Xuan Bui, and Jin-Cheol Kim. 2021. Nematicidal activity of cyclopiazonic acid derived from Penicillium commune against root-knot nematodes and optimization of the culture fermentation process. Front. Microbiol. 12:726504.
  • Yoon Jee Kim, Kalaiselvi Duraisamy, Min-Hye Jeong, Sook-Young Park, Soonok Kim, Yookyoung Lee, Van Thi Nguyen, Nan Hee Yu, Ae Ran Park, and Jin-Cheol Kim,*. 2021. Nematicidal activity of chemical intermediates of grammicin biosynthesis pathway in Xylaria grammica EL000614 against Meloidogyne incognita. Molecules 26(4675):1-13.
  • Ja Yeong Jang, Yong Ho Choi, Teak Soo Shin, Tae Hoon Kim, Kee-Sun Shin, Hae Woong Park, Young Ho Kim, Hun Kim, Gyung Ja Choi, Kyoung Soo Jang, Byeongjin Cha, In Seon Kim, Eul Jae Myung, and Jin-Cheol Kim*. 2016. Biological control of Meloidogyne incognita by Aspergillus niger F22 producing oxalic acid. PLOS ONE 11(6):e0156230.

Lines 62-66, 71: The scientific names should be written in italic.

The objectives of this study are not clear. Rewrite.

In the introduction section, the authors should explain why they evaluated the disease control efficacy of the two Bacillus strains against cucumber root-knot nematode.

Line 153: The production of gelatinase has been reported to be involved in nematicidal activity. Therefore the authors are required to test gelatinase activity of the two Bacillus strains.

Line 194: How did you know that only M. incognita can cause the cucumber root-knot nematode diseases in the greenhouse?

Lines 202-203: Rewrite.

Lines 212-213: Based on the number of galls and gall index, provide the control values of the two Bacillus strains.

What is the damage score? How did you get it?

Line 261: the nematicidal activity --> the disease control efficacy

Lines 264-265: Rewrite

Do in vitro nematicidal activity of the culture filtrates of the two Bacillus strains against second stage juveniles and eggs of M. incognita.

Line 296: a number of extracellular enzymes???? Are you sure? Did you test the productivity of a number of extracellular enzymes?

LInes 500-502:for the first time. Are you sure?

Conclusions: Rewrite.

Materials and Methods: Should be prepared in details.

References: There are so many errors?

Line 477: LC preparation. What does it mean?

How many fold were the samples diluted before their application?

Reviewer 2 Report

Manuscript entitled “Bacillus strains promising for protecting cucumber plants  against root-knot nematode Meloidogyne Incognita Kof. in a  greenhouse” fits within the journal scope.

I consider the presented results to be very interesting and valuable for the sciences of biological control of phytopathogenic nematodes. The Authors indicated the nematicidal activity of B. velezensis BZR86 and B. velezensis BZR277 against the root-knot nematode M. incognita of cucumber plants in greenhouse. These strains  enhance the plants resistance and decreased the numbers of the galls on the roots. In my opinion, the results obtained by Authors are promising, and Bacillus strains can be considered as antinematode agent to protection of cucumber plants.

After reading the manuscript, I have a few comments:

  1. Streptomyces is not fungus (see line 61). This is the Gram-positive bacteria that grows with a filamentous form similar to fungal hyphae.
  2. Latin names of organisms should always be written in italics. Please correct throughout the manuscript.
  3. In the abstract – “shoot mass and plant root (gall index) and productivity”. This part of the sentence is unclear. Please correct.
  4. Table 2: It is worth determining the significance of changes - as it was done in Tables 3 and 4.
  5. Line 181: “ food source” change to “nutrients source”
  6. Table 3 and 4: I suggest to use "a" for the highest value, and then the next letters of the alphabet according to the decreasing of the values. Such a presentation facilitates the analysis of data variability.
  7. Table 4: How the "damage score" was calculated? Please complete.
  8. Information on the role of chitinases in the elimination of nematodes should be added to the discussion. In my opinion, it should be mentioned that chitin is the essential component of nematode eggshell and pharynx. The chitin hydrolysis can lead to nematode embryonic lethal, laying defective eggs or moulting failure.
  9. Line 432: What parameter was an indicator of chitinase, lipase, and protease activity? How the activity of the enzymes was determined? I suppose there were specific zones around the colonies. Please complete the information. Readers unfamiliar with this subject will have to look for information in other literature positions.
  10. Lines 433-434: Was chitin added to the synthetic medium?
  11. Line 470-471: "on a potato-glucose medium, g/L: potato broth – 500.0" Is that the truth?
  12. Lines 476-477: “The number of nematodes was determined by Baermann method [58].” This is a very old reference. Whether the number of nematodes = the number of galls?  In the chapter “results” Authors described the gall index and number of galls. Please clarify.
  13. Lines 489-491: “The root gall severity was based on gall indices (GI) measured on 0 to 5 scales: 0-GI = 489 0%; 1-GI =10 to 20%; 2-GI = 21 to 50%; 3-GI = 51 to 80%; 4-GI = 81 to 100%., 5-GI = > 100% 490 on roots [59].” I do not find such data in the manuscript. The 5-point scale is only described in lines 489-491 (Materials and Methods).

According to me, this manuscript can be published after minor revision.

Reviewer 3 Report

This paper written by Anzhela M. Asaturova et al characterised and identified two Bacillus strains in vitro, and tested the effect of these two strains on root-knot nematode Meloidogyne Incognita Kof. of cucumber plants in a greenhouse condition. They found that these Bacillus can promote plant growth and productivity via inhibiting root-knot nematode in greenhouse via both single or double treatments.

The title looks grammatically incorrect, need a revision.

In the introduction, it’s better to start with more general information, such as the harmful effect of root-knot nematode on plants in general.

Too much information in Table 2, probably it’s better to just show the optimal growth conditions of these two strains in a table, and use figures to show the growth dynamics, five panels for these five tested growth conditions.

Are there any correlation between bacterial in vitro enzymatic activities and plant reactions in greenhouse? What the mechanism about the inhibition effect of bacterial strains on root-knot nematode?

In the M&M, authors wrote the “cultural and morphological characteristics of Bacillus strains”, but the results were not shown in the result section.

In general, there were too much information about these two strains themselves instead of their effect on root-knot nematode. I would suggest to shift the focus of the results.